# Effect of Obstructive Sleep Apnea on the Risk of Injuries—A Nationwide Population-Based Cohort Study

**DOI:** 10.3390/ijerph182413416

**Published:** 2021-12-20

**Authors:** An-Che Cheng, Gwo-Jang Wu, Chi-Hsiang Chung, Kuo-Hsiang Wu, Chien-An Sun, I-Duo Wang, Wu-Chien Chien

**Affiliations:** 1Department of Internal Medicine, Tri-Service General Hospital, National Defense Medical Center, Taipei City 11490, Taiwan; kwii781215@gmail.com; 2Graduate Institute of Medical Sciences, National Defense Medical Center, Taipei City 11490, Taiwan; gwojang@yahoo.com; 3Department of Obstetrics and Gynecology, Tri-Service General Hospital, National Defense Medical Center, Taipei City 11490, Taiwan; 4School of Public Health, National Defense Medical Center, Taipei City 11490, Taiwan; g694810042@gmail.com; 5Department of Medical Research, Tri-Service General Hospital, Taipei City 11490, Taiwan; 6Taiwanese Injury Prevention and Safety Promotion Association (TIPSPA), Taipei City 11490, Taiwan; 7Department of Nursing, Tri-Service General Hospital, National Defense Medical Center, Taipei City 11490, Taiwan; kamekame@mail.ndmctsgh.edu.tw; 8Big Data Research Center, College of Medicine, Fu-Jen Catholic University, New Taipei City 24205, Taiwan; 040866@mail.fju.edu.tw; 9Department of Public Health, College of Medicine, Fu-Jen Catholic University, New Taipei City 24205, Taiwan; 10Division of Pulmonary and Critical Care Medicine, Department of Internal Medicine, Tri-Service General Hospital, National Defense Medical Center, Taipei City 11490, Taiwan; 11Graduate Institute of Life Sciences, National Defense Medical Center, Taipei City 11490, Taiwan

**Keywords:** obstructive sleep apnoea, injuries, Taiwan national healths insurance research database, longitudinal health insurance database

## Abstract

Obstructive sleep apnea (OSA) has been reported to increase the risk of motor vehicle accidents. However, only few studies have investigated the effects of OSA on overall risk injury. The aim of study is to investigate whether OSA increases the risk of overall injury. The data were collected during 2000–2015 from Taiwan’s National Health Insurance Research Database. A total of 8901 individuals diagnosed with OSA were inpatients, or outpatients at least three times were enrolled. Finally, 6915 participants with OSA were included as the study cohort. We matched the study cohort with a comparison cohort, at a ratio of 1:4. Cox proportional hazards regression was used to analyse the association between OSA and overall injury. Patients with OSA had 83.1% increased risk of overall injury, compared to non-OSA individuals [adjusted hazards ratio (HR) = 1.831, confidence interval (CI) = 1.674–2.020, *p* < 0.001]. In the stratified age group, patients aged ≧65 years had the highest risk of injury (adjusted HR= 2.014; CI = 1.842–2.222, *p* < 0.001). Patients with OSA were at a higher risk of falls, traffic injury, poisoning, suffocation, suicide, and abuse or homicide than non-OSA individuals, with falls and traffic injury as the leading causes of injuries. The data demonstrated that patients with OSA have a higher risk of overall injury. The study results can be a reference for developing injury prevention strategies in the future. The general population and clinicians should have more awareness regarding OSA and its negative effects on injury development.

## 1. Introduction

Obstructive sleep apnoea (OSA) is a prevalent, yet under-diagnosed respiratory disorder, characterised by frequent episodes of upper airway collapse during sleep. Approximately 1 billion adult population have sleep apnoea [1]. OSA causes sleep fragmentation and repetitive hypoxemia and is associated with various adverse outcomes, including daytime sleepiness, decreased learning skills, and impaired neurocognitive function [2]. OSA has been strongly associated with long-term health consequences, including hypertension, cardiovascular events, cerebrovascular disease, diabetes, and depression [3,4,5,6].

The relationship between OSA and motor vehicle accidents (MVAs) has been well-established [7,8] and become one of the top priorities of the Federal Motor Carrier Safety Administration in the United States. However, only few studies have investigated the relationship between OSA and its effect on injury development in the general adult population. In the present study, we used data from Taiwan’s National Health Insurance Research Database (NHIRD), a longitudinal nationwide population-based cohort study, with data of follow-up for more than 16 years, with an aim to determine whether OSA is associated with the risk of overall injury.

## 2. Material and Methods

### 2.1. Data Sources

We used data from the National Health Insurance Research Database (NHIRD) in Taiwan. Taiwan has started the National Health Insurance (NHI) program, since 1995. It includes the data of approximately 99% of the people in Taiwan (a total of 23.74 million people) and 97% of the clinics are covered under the NHI system [9]. The NHIRD contains several health registration records of the Taiwanese general population, including outpatient, inpatient, and emergency department data. Physicians coded the diagnosis according to the International Classification of Diseases, Ninth Revision, Clinical Modification (ICD-9-CM). The NHIRD has high accuracy and validity, in terms of diagnoses [10]. Therefore, the NHIRD offers representative data for medical- and health-related research. Moreover, the NHIRD encrypts and converts all the identification numbers in all NHIRD records to protect the privacy of all people registered in the program. Our study protocol was approved by the Institutional Review Board of Tri-Service General Hospital (TSGHIRB No. B-110-55).

### 2.2. Study Design and Participants

This population-based cohort study enrolled 1,936,512 outpatients and inpatients from the Longitudinal Health Insurance Database, between 1 January 2000 and 31 December 2015, in Taiwan. We included patients aged above 20 years who were diagnosed with OSA (ICD-9-CM code: 780.51, 780.53, 780.57) and had at least three outpatient visits or one hospitalization among the 8901 individuals. OSA is diagnosed by otorhinolaryngologists and pulmonologists, using overnight polysomnography (PSG) to detect the frequency of apnoeic and hypopnoeic events. The apnoea–hypopnoea index (AHI) is the average number of disordered breathing events per hour. OSA syndrome is defined as an AHI of ≥5 associated symptoms (e.g., excessive daytime sleepiness, fatigue, or impaired cognition) or an AHI of ≥15, regardless of associated symptoms [11]. Patients diagnosed with OSA and those who had injuries before 1 January 2000 were excluded. The index date was defined as the date of a new diagnosis of OSA. In total, 6915 participants who matched our inclusion criteria were assigned to the study cohort. Individuals without OSA diagnosis were randomly selected for the control group and matched with the OSA group by age, sex, index date, and comorbidities, using a four-fold propensity score matching. Among the control patients, we excluded those with a history of sleep disorders in the study period and with a medical history, following the exclusion criteria. The flowchart of the study design is shown in Figure 1.

### 2.3. Outcome and Comorbidity

All the participants were followed from the index date until the onset of injury or the end of 31 December 2015. Different categories of injury according to the ICD-9 diagnosis, such as fracture, dislocation, sprain and strains, intracranial/internal injury, open wound, injury to blood vessels, superficial injury, contusion, crushing, foreign body entering through orifice, burn, injury to nerves and spinal cord, poisoning and others, were included. We categorised the mechanisms of injuries according to the ICD-9 diagnosis, including traffic injury, falls, suffocation, other unintentional injuries, and suicide. Common comorbidities such as diabetes mellitus (DM), hypertension (HTN), hyperlipidaemia, cerebrovascular disease (CVD), stroke, obesity, anxiety, and depression were listed and compared the baseline characteristics between the two groups. The ICD-9-CM codes used in the present study are shown in Appendix A.

### 2.4. Statistical Analysis

The demographic features and common comorbidities between patients with OSA and non-OSA individuals were compared using the chi-square test. Mean patient age in both cohorts was calculated using Student’s t-test. The incidence rate (per 10^5^ person-years) of injury was calculated according to sex, age, and comorbidities for each cohort. Age, sex, and concomitant comorbidities were included in the multivariable model for adjustment. Hazard ratios (HRs) and 95% confidence intervals (CIs) were calculated using multivariable Cox proportional hazards models. We evaluated the cumulative incidence of OSA-related injury using the Kaplan–Meier analysis and log-rank test. All statistical analyses were conducted using SPSS 22.0 software for Windows, version 22.0 (IBM Corp., Armonk, NY, USA), with a two-tailed test *p* value of <0.05 for statistical significance.

## 3. Results

### 3.1. Baseline Characteristics of the Patients in the Study

The study included 6915 patients (mean [SD] age, 56.11[17.94] years) in the OSA cohort and 12,100 patients (mean [SD] age, 56.14[18.04] years) in the control cohort, 2000–2015 (Figure 1). Data of baseline demographic characteristics and common comorbidities of patients are shown in Appendix A. Most patients in our study population were male (67.10%). Among the study population, 39.81% of patients were aged more than 65 years, 35.23% of patients were aged 45–64 years, and 24.96% of patients were aged 20–44 years. The distribution of age, sex, and common comorbidities was similar between the study and control cohorts.

### 3.2. Endpoint Characteristics of the Study

Information of the endpoint characteristics is presented in Table 1. At the end of the follow-up, injury events developed in 1770 patients in the OSA cohort and 6675 patients in the control cohort (Figure 1). The mean follow-up time was 10.29 ± 10.76 years in the OSA cohort and 10.94 ± 11.29 in the control cohort (Appendix A). The mean number of years to injury was 5.54 ± 4.22 in OSA cohort and 5.63 ± 4.33 in the control cohort (Appendix A). The prevalence of DM, HTN, stroke, and obesity in OSA cohort were higher than control cohort (all *p* < 0.001).

### 3.3. Comparison between the Incidence and Risk of Injury of the OSA and Control Cohorts

We stratified and analysed sex, age, and concomitant comorbidities for the risk of injury using Cox regression model as presented in Table 2. After adjusting for age, sex, and other concomitant comorbidities, patients with OSA had an 83.1% higher risk of injury (adjusted HR = 1.831 [95% CI, 1.674–2.202]; *p*< 0.001) than non-OSA individuals. Patients with OSA had a higher incidence of injury than the non-OSA individuals (2590 vs. 2278 per 10^5^ person-years, *p* < 0.05). The Kaplan–Meier curve showed that the incidence of injury started to branch off in the first year in the OSA group and persisted until the end of the follow-up (log-rank; *p* < 0.001; Figure 2).

In the analysis of stratified age groups, OSA had the greatest impact on the age group of more than 65 years (adjusted HR = 2.014 [95% CI, 1.842–2.222]; *p* < 0.001), followed by the age group of 45–64 (adjusted HR = 1.746 [95% CI, 1.596–1.926]; *p* < 0.001) and the age group of 20–44 years (adjusted HR = 1.741 [95% CI, 1.592–1.921]; *p* < 0.001) (Table 2). The risk of injury in patients with OSA increased with concomitant comorbidities, including DM, HTN, hyperlipidaemia, CVD, stroke, obesity, anxiety, and depression (all *p* < 0.001).

### 3.4. Diagnosis and Causes of Injury

The subgroup analysis of injury, including its diagnosis and causes, through Cox regression, are presented in Table 3. For the diagnosis, patients with OSA had a higher risk of fracture, dislocation, intracranial injury, open wound, blood vessel injury, contusion and crushing injury, foreign body entering through an orifice, nerve and spinal cord injury, poisoning, and other injuries than non-OSA individuals.

For the causes of injury, falls had the highest incidence (490.26 injuries per 10^5^ person-years), followed by traffic injury (440.50 injuries per 10^5^ person-years). In unintentional injury, patients with OSA had higher risk for falls [adjusted HR = 1.564 (95% CI, 1.430–1.726); *p* < 0.001], traffic injury [adjusted HR = 1.808 (95% CI, 1.653–1.994); *p* < 0.001], poisoning (solid and liquid substances) [adjusted HR = 1.752 (95% CI, 1.068–1.933); *p* < 0.001], and suffocation [adjusted HR = 1.168 (95% CI, 1.602–1.289); *p* < 0.001] than the non-OSA individuals. In intentional injury, patients with OSA had a higher risk of suicide by 6.683-fold [adjusted HR = 6.683 (95% CI, 6.110–7.372); *p* < 0.001] and homicide or abuse by 10.888-fold [adjusted HR = 10.888 (95% CI, 9.955–12.012); *p* < 0.001] than the non-OSA individuals.

## 4. Discussion

This population-based cohort study demonstrated that patients with OSA had a higher incidence rate and 83.1% higher risk of overall injury than those without OSA. In the stratified age group, the older patients (≥65 years old) with OSA had a 2.01-fold greater risk of injury than those without OSA. Patients with OSA were at a higher risk of unintentional injuries, including falls and substance poisoning, traffic injury, and suffocation, than non-OSA individuals. In intentional injury, patients with OSA had a 6.68- and 10.88-fold greater risk of suicide and abuse or homicide-related injury, respectively, than non-OSA individuals.

Past studies [7,8] have reported that patients with OSA had a higher risk of developing MVA and crash-related injury than non-OSA individuals. Moreover, a strong relationship has been reported between OSA and the risk of occupational injury [2,12,13]. However, investigations on the relationship between OSA and its effect on injury development in the general population are limited. To the best of our knowledge, our study is the largest and longest nationwide cohort study that investigated the association between OSA and the risk of overall injury.

For the unintentional injuries, falls had the highest incidence, followed by traffic injury. The results were similar to previous observations [14,15], showing traffic injury and falls as the prevalent leading causes of unintentional injury in Taiwan, especially in the older population. For the risk of unintentional injury subtypes, patients with OSA had higher risks of falls, traffic injury, poisoning (solid and liquid substances), suffocation, and other unintentional injuries than non-OSA individuals. Our study provided information on the risk of different unintentional injury types in patients with OSA, which could be used as a reference for establishing injury prevention strategies in future studies.

The mechanism of OSA, and its association with different types of unintentional injuries, are not completely clear. There are several possible explanations for these findings. First, previous studies suggest neurocognitive defects in patients with OSA, including decline in cognition, attention, memory, vigilance, executive functioning, and even visuospatial/constructional abilities [16,17,18,19]. These may cause overestimation of their ability, misjudged environmental hazards, mistaken identification of inedible substances, such as disinfectants as medications, and ingestion of foreign bodies. Second, patients with OSA commonly present with poor sleep quality, nocturnal hypoxemia, and excessive daytime sleepiness [20]. Furdato et al. [21] demonstrated that healthy individuals with long-term poor sleep quality have worsened static and dynamic postural stability, compared to those with good sleep quality. Degache et al. [22] examined that the nocturnal oxygen saturation level has the strongest independent association with impaired daytime postural control. Sleep loss status has a negative effect on the driver’s maintenance of lane positions, steering in simulation, and reaction time [23]. Excessive daytime sleepiness also contributes to MVAs [7,24] and an increase in the risk of falls, especially in females [25,26]. Third, a previous study demonstrated that patients with OSA were likely to present with impaired swallowing reflex [27]. Another study [28] examined that symptomatic patients with OSA had an increased risk of pharyngeal aspiration than the normal control group. This mechanism is still unclear, but it may be due to the perturbed neural and muscular function of the upper airways [27]; thus, it may be associated with an increased risk of suffocation. Further investigations are needed for understanding OSA and its effect on different types of injuries.

The relationship between sleep apnoea and psychiatric disorders, including mood, anxiety, depression, and schizophrenia, is well-established [29,30,31]. For example, a previous study also demonstrated a positive association between OSA and suicidal thinking [32]. However, there are limited data comparing the risk of suicide-related injury between patients with and without OSA. Our study reported that OSA was significantly associated with suicide-related injury. Nevertheless, a recent large national sample study [33] found that sleep apnoea was not significantly associated with past-year suicide attempts, which was not consistent with the findings of our study. This inconsistency, assessed using single self-report items, may be due to many variables, including suicidal ideation, planning, and attempt. Thus, the methodology may have led to the underestimation of suicide prevalence. The present study demonstrated that OSA should be considered a risk factor for suicide-related injury. Moreover, the early diagnosis of OSA may provide clinicians the opportunity to discuss mental health with their patients, resulting in early prevention of suicide-related injury.

Although the specific mechanism of OSA in suicidality remains unclear, several hypotheses have been proposed to explain this relationship. For example, sleep disruption affected mental health, including increased impulsivity and irritability, difficulty in maintaining motivation, and emotional dysregulation [34], all of which may impair the patient’s ability to avoid suicidal thought and behaviour. Furthermore, chronic disruption of the sleep cycle can decrease mental and physical energy, potentially affecting the ability to deal with stress and mental difficulties [35]. Recent literature [36,37,38] revealed that OSA is associated with the development of many comorbidities, such as cardiovascular, cerebrovascular, and metabolic diseases, all of which may lead to decreased quality of life, indirectly increasing suicide risk. Future research is needed to investigate the mechanisms, as well as the direct and indirect effects of OSA on mental health.

Another important finding in the present study was that patients with OSA were 10.88 times more likely to be injured due to homicide or abuse than those without OSA, especially women, who are associated with a higher risk than men with OSA. To the best of our knowledge, this is the first study to prove that OSA is a novel risk factor for adult abuse or homicide-related injury; however, the underlying reasons have yet to be elucidated. This result may be because patients with OSA had a higher prevalence of comorbidities [36,37,38], such as cardiovascular and neurocognitive dysfunction [18,19] and psychiatric disorders [30,31], than non-OSA individuals. As mentioned above, these were the risk factors for abuse or homicide victimization [39,40]. Therefore, they are more likely to become a victim of abuse or homicide than those without OSA. Moreover, snoring could impair the sleep and life quality of their bed partner or caregivers [41] and lead to frustration, irritation, exhaustion, depression, strained relationship, and marital dissatisfaction [42,43]. In some cases, this may result in domestic violence [44], which was accounts for approximately 40% of homicide cases in the U.S. [45]. Abuse may be emotional, physical, or sexual. However, we were unable to conduct a detailed analysis on abuse because the database contained no information on its various types. Further research should be conducted to better understand the relationship between OSA and the type of abuse and underlying mechanisms. Subsequently, this will facilitate the development of abuse prevention strategies, practices, and policies.

The present study showed that the increased incidence and risk in injury was most pronounced among elderly patients with OSA aged more than 65 years, followed by those aged 45–64 years and 20–44 years. These results suggest that the effects of OSA on injury development were greater in elderly patients than in younger patients. There are several possible explanations for this result. A recent study [46] compared clinical and polysomnographic characteristics of patients with OSA above and below 65 years of age. The results revealed that the apnoea–hypopnea index was higher, whereas mean oxygen saturation was lower in elderly patients than in younger patients. All of the above findings may be associated with poorer sleep quality, intermittent chronic hypoxia, and increased daytime sleepiness in older patients. Poor sleep quality may have impaired postural control and stability [21,22]. Chronic hypoxia may affect brain functions, such as attention, vigilance, and coordination [16,47,48]. Moreover, excessive daytime sleepiness is associated with at least one event of fall in one year, especially in elderly female [26]. The result of the current study suggests that clinicians should pay more attention to the effect of OSA on injury development and implement injury preventive measures for older patients with OSA.

This study has some limitations. First, the polysomnographic data were not available. We could not associate the disease severity with injury risk. There was a lack of data on the degree of excessive daytime sleepiness, hypoxemia, sleep fragmentation, and sleep quality, which limited the generalisation of the results. Second, no comprehensive information on patient characteristics was recorded in the Taiwan’s NHIRD, such as on lifestyle, behaviour pattern, alcohol consumption, psychosocial and environmental factors, and medications. Patients with OSA had a higher prevalence of cardiovascular [36,37,38] and psychiatric disorders [30,31] than non-OSA individuals. They may be prescribed with anti-depressive and anti-hypertensive agents, which may have caused dizziness [49,50], hyper somnolence [50,51], or cognitive dysfunction [52]; however, we could not confirm the drug effect on the risk of injury. Third, only those who have symptoms and sought medical service at clinics or emergency rooms were included in the database because many patients with OSA are asymptomatic [53]. Thus, that may have caused us to underestimate the OSA prevalence or injury incidence, and we could not exclude the individuals without OSA diagnosis in the control group who were actually victims of OSA. Fourth, OSA and insomnia are two prevalent sleep disorders, and they often coexist interacting to amplify an overall more severe illness [54,55,56]. A recent study reported the prevalence of comorbid insomnia and OSA (COMISA); 35% of insomnia patients have an AHI of ≥5, and 29% have an AHI of ≥15, while 38% of patients with OSA meet insomnia criteria [57]. However, in our study, we could not exclude the patients with COMISA or those who are comorbid, with other sleep disorders. Fifth, previous studies demonstrated continuous positive airway pressure (CPAP) treatment and surgical intervention could reduce the risk of MVA [58], improve cognitive dysfunction [59], and reduce daytime sleepiness [60] in patients with OSA, which might have resulted in a decreased injury risk. However, we could not exclude the patients with OSA who accepted CPAP or surgery. This is a potential confounding factor, which may have affected the study results.

## 5. Conclusions

OSA was independently associated with an increased overall injury risk. The effect of OSA on the risk of injury increases with age. Patients with OSA aged >65 years old, 45–64 years old, and 45–64 years old were 2.014, 1.746, and 1.741, respectively, with -fold greater risk of injury than those without OSA. For the different causes of injury, patients with OSA were at a higher risk of falls and substances poisoning, traffic injury, suffocation, suicide-related injury, and abuse- or homicide-related injury. The results of this study may serve as a reference for the development of injury prevention strategies. Moreover, the general population and clinicians should have more awareness, regarding OSA and its negative effects on injury development.

## Figures and Tables

**Figure 1 ijerph-18-13416-f001:**
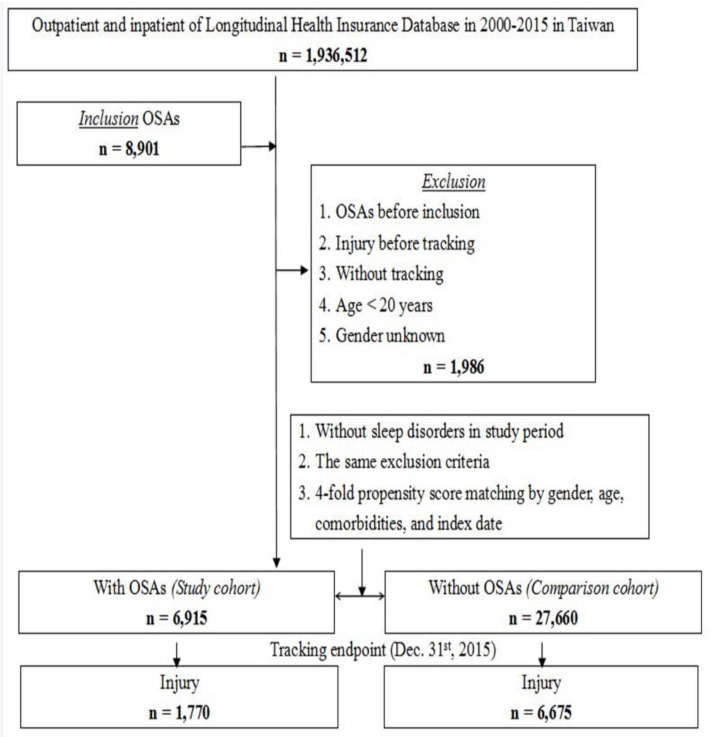
Flowchart of study group selection from the National Health Insurance Research Database in Taiwan.

**Figure 2 ijerph-18-13416-f002:**
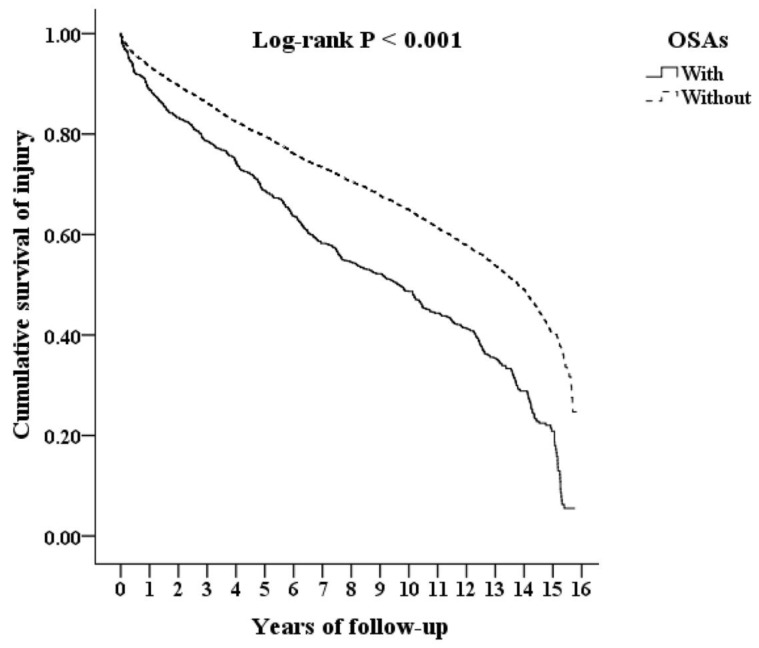
Kaplan-Meier for cumulative survive of injury among aged 20 and over stratified by OSAs with log-rank test.

**Table 1 ijerph-18-13416-t001:** Endpoint characteristics of the study.

OSAs	Total		With		Without		*P*
Variables	*n*	%	*n*	%	*n*	%	
**Total**	34,575		6915	20.00	27,660	80.00	
**Injury**							0.011
Without	26,130	75.57	5145	74.40	20,985	75.87	
With	8445	24.43	1770	25.6	6675	24.13	
**Gender**							0.999
Male	23,200	67.10	4640	67.10	18,560	67.10	
Female	11,375	32.90	2275	32.90	9100	32.90	
**Age (y)**	66.87 ± 19.48		66.04 ± 18.26		67.08 ± 19.77		<0.001
**Age group (y)**							<0.001
20–44	6871	19.87	1529	22.11	5342	19.31	
45–64	11,746	33.97	2442	35.31	9304	33.64	
≧65	15,958	46.15	2944	42.57	13,014	47.05	
**DM**							<0.001
Without	28,993	83.86	5573	80.59	23,420	84.67	
With	5582	16.14	1342	19.41	4240	15.33	
**HTN**							<0.001
Without	28,247	81.70	5442	78.70	22,805	82.45	
With	6328	18.30	1473	21.30	4855	17.55	
**Hyperlipi** **demia**							0.208
Without	33,895	98.03	6766	97.85	27,129	98.08	
With	680	1.97	149	2.15	531	1.92	
**CVD**							0.399
Without	31,677	91.62	6318	91.37	25,359	91.68	
With	2898	8.38	597	8.63	2301	8.32	
**Stroke**							<0.001
Without	31,761	91.86	6231	90.11	25,530	92.30	
With	2814	8.14	684	9.89	2130	7.70	
**Obesity**							<0.001
Without	34,550	99.93	6900	99.78	27,650	99.96	
With	25	0.07	15	0.22	10	0.04	
**Anxiety**							0.001
Without	34,455	99.65	6876	99.44	27,579	99.71	
With	120	0.35	39	0.56	81	0.29	
**Depression**							0.004
Without	34,336	99.31	6849	99.05	27,487	99.37	
With	239	0.69	66	0.95	173	0.63	

*P*: Chi-square/Fisher exact test on category variables and t-test on continue variables. Abbreviations: OSA = obstructive sleep apnea; y = years; DM = diabetes mellitus;.HTN = hypertension; CVD = cardiovascular disease.

**Table 2 ijerph-18-13416-t002:** Injury factors stratified by variables by using Cox regression.

OSA	With			Without			With vs. Without				
Stratified	Events	PYs	Rate (per 10^5^ PYs)	Events	PYs	Rate (per 10^5^ PYs)	Ratio	Adjusted HR	95% CI	95% CI	*P*
**Total**	1770	68,331.70	2590.31	6775	297,346.61	2278.49	1.137	1.831	1.674	2.020	<0.001
**Gender**											
Male	1162	43,533.39	2669.22	4077	173,185.76	2354.12	1.134	1.826	1.670	2.015	<0.001
Female	608	24,798.31	2451.78	2698	124,160.85	2172.99	1.128	1.817	1.661	2.005	<0.001
**Age group (y)**											
20-44	298	14,504.71	2054.51	797	41,937.46	1900.45	1.081	1.741	1.592	1.921	<0.001
45-64	730	28,126.24	2595.44	2284	95,380.39	2394.62	1.084	1.746	1.596	1.926	<0.001
≧65	742	25,700.75	2887.07	3694	160,028.75	2308.34	1.251	2.014	1.842	2.222	<0.001
**DM**											
Without	1491	57,447.27	2595.42	5533	240,331.50	2302.24	1.127	1.816	1.660	2.003	<0.001
With	279	10,884.43	2563.29	1242	57,015.11	2178.37	1.177	1.895	1.733	2.091	<0.001
**HTN**											
Without	1435	52,024.02	2758.34	5430	221,961.83	2446.37	1.128	1.816	1.660	2.003	<0.001
With	335	16,307.68	2054.25	1345	75,384.78	1784.18	1.151	1.854	1.695	2.046	<0.001
**Hyperlipi** **demia**											
Without	1711	65,627.41	2607.14	6714	288,258.77	2329.16	1.119	1.803	1.648	1.989	<0.001
With	59	2704.30	2181.71	61	9087.84	671.23	3.250	5.235	4.786	5.775	<0.001
**CVD**											
Without	1642	61,274.03	2679.76	6321	266,420.25	2372.57	1.129	1.819	1.663	2.007	<0.001
With	128	7057.67	1813.63	454	30,926.35	1468.00	1.235	1.990	1.819	2.195	<0.001
**Stroke**											
Without	1659	62,945.26	2635.62	6393	273,043.04	2341.39	1.126	1.813	1.658	2.000	<0.001
With	111	5386.45	2060.73	382	24,303.57	1571.79	1.311	2.112	1.931	2.330	<0.001
**Obesity**											
Without	1767	68,113.39	2594.20	6772	297,084.57	2279.49	1.138	1.833	1.676	2.022	<0.001
With	3	218.31	1374.20	3	262.03	1144.89	1.200	1.933	1.767	2.133	<0.001
**Anxiety**											
Without	1763	68,218.88	2584.33	6764	296,229.62	2283.36	1.132	1.823	1.667	2.011	<0.001
With	7	112.82	6204.42	11	1,116.98	984.80	6.300	10.147	9.277	11.194	<0.001
**Depression**											
Without	1747	67,546.47	2586.37	6718	294,942.01	2277.74	1.135	1.829	1.672	2.018	<0.001
With	23	785.23	2929.06	57	2404.59	2370.46	1.236	1.990	1.819	2.196	<0.001

Abbreviations: OSA = obstructive sleep apnea; y = years; PYs = person-years; Adjusted HR = adjusted hazard ratio: adjusted for the variables listed in Table 2; CI = confidence interval.

**Table 3 ijerph-18-13416-t003:** Factors of injury subgroup by using Cox regression.

OSA	With			Without			With vs. Without				
Injury Subgroup	Events	PYs	Rate (per 10^5^ PYs)	Events	PYs	Rate (per 10^5^ PYs)	Ratio	Adjusted HR	95% CI	95% CI	*p*
**Injury diagnosis**											
Fracture	513	68,331.70	750.75	2321	297,346.61	780.57	0.962	1.549	1.416	1.709	<0.001
Dislocation	25	68,331.70	36.59	120	297,346.61	40.36	0.907	1.460	1.335	1.611	<0.001
Sprains and strains	18	68,331.70	26.34	153	297,346.61	51.46	0.512	0.825	0.754	0.910	<0.001
Intracranial / internal injury	273	68,331.70	399.52	1,017	297,346.61	342.03	1.168	1.881	1.720	2.076	<0.001
Open wound	134	68,331.70	196.10	510	297,346.61	171.52	1.143	1.841	1.684	2.032	<0.001
Injury to blood vessels	16	68,331.70	23.42	9	297,346.61	3.03	7.736	12.460	11.391	13.746	<0.001
Superficial injury / contusion	55	68,331.70	80.49	276	297,346.61	92.82	0.867	1.397	1.277	1.541	<0.001
Crushing	18	68,331.70	26.34	32	297,346.61	10.76	2.448	3.942	3.604	4.349	<0.001
Foreign body entering through orifice	9	68,331.70	13.17	35	297,346.61	11.77	1.119	1.802	1.648	1.988	<0.001
Injury to nerves and spinal cord	5	68,331.70	7.32	32	297,346.61	10.76	0.680	1.095	1.001	1.208	0.049
Poisoning	130	68,331.70	190.25	347	297,346.61	116.70	1.630	2.626	2.401	2.897	<0.001
Others injury	567	68,331.70	829.78	1819	297,346.61	611.74	1.356	2.185	1.997	2.410	<0.001
**Cause of injury**											
Traffic	301	68,331.70	440.50	1167	297,346.61	392.47	1.122	1.808	1.653	1.994	<0.001
Poisoning (solid and liquid substances )	7	68,331.70	10.24	28	297,346.61	9.42	1.088	1.752	1.602	1.933	<0.001
Falls	335	68,331.70	490.26	1501	297,346.61	504.80	0.971	1.564	1.430	1.726	<0.001
Suffocation	5	68,331.70	7.32	30	297,346.61	10.09	0.725	1.168	1.068	1.289	<0.001
Suicide	82	68,331.70	120.00	86	297,346.61	28.92	4.149	6.683	6.110	7.372	<0.001
Homicide/Abuse	87	68,331.70	127.32	56	297,346.61	18.83	6.760	10.888	9.955	12.012	<0.001
**Intentionality** **of injury**											
Unintentional	1124	68,331.70	1644.92	4285	297,346.61	1441.08	1.141	1.838	1.681	2.028	<0.001
Intentional	169	68,331.70	247.32	142	297,346.61	47.76	5.179	8.341	7.626	9.202	<0.001

Abbreviations: OSA = obstructive sleep apnea; y = years; PYs = person-years; Adjusted HR = adjusted hazard ratio: adjusted for sex, age, comorbidities (diabetes mellitus, hypertension, hyperlipidemia, cerebrovascular disease, stroke, obesity, anxiety, depression); CI = confidence interval. Unintentional: traffic; poisoning (solid and liquid substances); falls; suffocation. Intentional: suicide; homicide/abuse.

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
