# Peer review of "Effect of Obstructive Sleep Apnea on the Risk of Injuries—A Nationwide Population-Based Cohort Study"

_ijerph, 2021, doi:10.3390/ijerph182413416_

Round 1

Reviewer 1 Report

The paper "Effect of Obstructive Sleep Apnea on the Risk of Injuries- A 2
Nationwide Population-Based Cohort Study" by An-Che Cheng et al. provides interested data about a large variety of possible injuries in patients with OSA, followed for over 5 years. It is well structured, with a solid statistical analysis. The results are original. There are some issues and questions that can be answered. 

  1. Page 1. Introduction, reference 1. You can add another one with the updated information: approx. 1 billion persons suffer of sleep apnea.
  2. Page 2. line 79. "At least three outpatient visits or one hospitalisation". Do you have data about the treatment impact? Some medication might cause hyper somnolence, dizziness, coordination deficit, etc. 
  3. Page 4. Results. Data about the CPAP treatment? How about the impact of treatment on cognitive function and risk of injuries?
  4. Page 10, paragraph "In addition, several studies reported that patients with OSA had a higher prevalence of comorbidities, such as hypertension, ischemic stroke, diabetes [41-43], neurocognitive dysfunction [18, 25, 9], and psychiatric disorders, including bipolar disorder, anxiety, and depression [30, 31]than non-OSA individuals " it it is repeated. 
  5. Page 11. Conclusion. Elderly with comorbidities are frail with high risk of different injuries. In my opinion you need to show the degree of risk in younger patients with OSA. 

Author Response

We are deeply honored by the effort you spent in reviewing this manuscript, and greatly appreciate the positive comments from you. In reviewing and revising our text, we are motivated to read more and learn more from your criticisms. Our point by point responses to your comments are listed below and highlighted the changes in manuscript (red color).

Reviewer 2 Report

I have read the article by Cheng et al. with great interest. The authors investigated the risk of injuries in patients with OSA in a population-based study. This is truly an interesting and large study, however due to the critical concerns mentioned below, I cannot support the publication of this paper.

Comments:

  • OSA is a frequent disease which has a prevalence of 30-50% in the adult population. https://pubmed.ncbi.nlm.nih.gov/31300334/. It is not clear, how was OSA excluded in the control group. My feeling is that was based on symptoms, but the majority of patients with OSA do not have daytime symptoms.
  • What diagnostic criteria were used to diagnose OSA?
  • Once, patients were diagnosed, I presume some of them were treated. There is some evidence that CPAP treatment reduces the number of accidents. Yet, this important covariate was not considered in the analysis.
  • As discussed, excessive daytime sleepiness could be the most likely reason for the accidents. Do you have any data on this? During the diagnosis, clearly patients should have had this assessed.

Author Response

(The authors gave the same response as above.)

Reviewer 3 Report

This study investigates on the risk of motor vehicle accidents of people with Obstructive sleep apnea (OSA). To this aim, the authors considered data from Taiwan's National Health Insurance Research Database. A final cohort of 6915 patients was matched to 27660 controls (ratio of 1:4).

As main findings, patients with OSA had

  • increased risk of overall injury compared to non-OSA individuals
  • this risk was higher in aged >65 years patients
  • the overall risk of injury was associated with (in the order): falls, traffic injury, poisoning, suffocation, suicide, and abuse or homicide than non-OSA individuals

Although not completely novel, I have a positive evalation on this study. It is the largest nationwide cohort study investigating on the association between OSA and the risk of overall injury.

Obviously, it has the intrinsic limitaions of these kind of studies, and any explanation of the reported differences remains speculative, due to its correlational nature.

I have only very minor points:

  • please, report in the abstract the final cohort of 6915 participants
  • the presence of other sleep disorders was an exclusion criterion. Please, clarify how these sleep disorders were detected. This is an important issue due to the extent and/or relevance of Co-Morbid Insomnia in OSAs patients [De Gennaro L. The State of Art on Co-Morbid Insomnia and Sleep Apnea (COMISA). Brain Sciences. 2021; 11(8):1079. https://doi.org/10.3390/brainsci11081079]
  • I do not know if the Taiwan's National Health Insurance Research Database provides any information on treatments. It is a quite relevant point and some information on the modulating role of treatments seems crucial

Author Response

(The authors gave the same response as above.)

Round 2

Reviewer 2 Report

I am glad that the authors acknowledged all limitations, however I feel they are too essential to support publication of this study.

Author Response

We are deeply grateful for the effort you spent in reviewing this manuscript, and greatly appreciate the comments from you. Due to methodological limitations, there are potential confounding factors in the study. Actually, we are studying the association between CPAP treatments in patients with OSA and different types of injury risk. We will conduct the study and consider the important covariates you recommend previously.